# mTOR and Aging: An Old Fashioned Dress

**DOI:** 10.3390/ijms20112774

**Published:** 2019-06-06

**Authors:** Giovanni Stallone, Barbara Infante, Concetta Prisciandaro, Giuseppe Grandaliano

**Affiliations:** Department of Medical and Surgical Sciences, Nephrology, Dialysis and Transplantation Unit, University of Foggia, Viale Luigi Pinto, 1, 71100 Foggia, Italy; barbarinf@libero.it (B.I.); concetta.prisciandaro@gmail.com (C.P.)

**Keywords:** mTOR, aging, cardiovascular system, immune system, cancer

## Abstract

Aging is a physiologic/pathologic process characterized by a progressive impairment of cellular functions, supported by the alterations of several molecular pathways, leading to an increased cell susceptibility to injury. This deterioration is the primary risk factor for several major human pathologies. Numerous cellular processes, including genomic instability, telomere erosion, epigenetic alterations, loss of proteostasis, deregulated nutrient-sensing, mitochondrial dysfunction, stem cell exhaustion, and altered intercellular signal transduction represent common denominators of aging in different organisms. Mammalian target of rapamycin (mTOR) is an evolutionarily conserved nutrient sensing protein kinase that regulates growth and metabolism in all eukaryotic cells. Studies in flies, worms, yeast, and mice support the hypothesis that the mTOR signalling network plays a pivotal role in modulating aging. mTOR is emerging as the most robust mediator of the protective effects of various forms of dietary restriction, which has been shown to extend lifespan and slow the onset of age-related diseases across species. Herein we discuss the role of mTor signalling network in the development of classic age-related diseases, focused on cardiovascular system, immune response, and cancer.

## 1. Introduction

Aging is a time-dependent physiological functional decline that affects most living organisms, supported by alterations of several molecular pathways, and represents the most profound risk factor for many non-communicable diseases [1].

Several studies suggest that aging should be defined as a disease [2,3,4,5]. In particular, recently, Blagosklonny defined aging as a disease on the basis of potentially available specific treatments [6]. On the other hand, Dilman, more than thirty years ago, described the aging process as a “normal disease” [7,8]. Summarising these notions, we could be in agreement with those who consider aging as “the sum of all age-related diseases” or a key “pre-disease” [9]. Indeed, the functional deterioration featuring aging is a primary risk factor for major human pathologies, including cancer, diabetes, cardiovascular disorders, and neurodegenerative diseases.

Several cellular processes, including genomic instability, telomere attrition, epigenetic alterations, loss of proteostasis, deregulated nutrient-sensing, mitochondrial dysfunction, cellular senescence, stem cell exhaustion, and altered intercellular-communication, represent common denominators of aging in different organisms. In this scenario Mammalian target of rapamycin (mTOR) has been suggested to play a pivotal role [10].

mTOR is a serine/threonine kinase, which belongs to the phosphoinositide kinase-related family. It is a functional enzyme that can be found within two distinct complexes: mTOR complex1 (mTORC1) and mTORC2, both characterized by different protein partners and specific substrates [10]. mTORC1 is activated by several growth factors through phosphoinositide-3-(PI3)-kinase-related kinase family and AKT kinase signaling. mTORC1 is also activated by nutrients (i.e., amino acids, phosphates) and repressed by AMP-activated protein kinase (AMPK), a key sensor of cellular energy status. Therefore, it is involved in cell growth, proliferation, survival, motility, autophagy, and protein synthesis [10]. This mechanism is one of the best-characterized pathways of lifespan. In fact, low levels of insulin and IGF-1, two growth factors activating mTOR, induced by caloric restriction are associated with an health span improvement and longevity increase. mTORC2, generally less sensitive to rapamycin and inhibited by chronic treatment, plays a crucial role in metabolic control but also in spatial control through the actin cytoskeleton regulation [11,12]. Interestingly, several observations in different animal models suggest that long-term inhibition of mTOR is associated with a significant extension of lifespan [13,14,15,16].

## 2. Protein Translation

mTORC1 regulates several key steps of protein synthesis, controlling the expression of genes promoting cell proliferation and survival. Protein synthesis requires not only functional ribosomes but also the coordinated activity of several translation initiation and elongation factors [17,18]. Two key targets of mTORC1 signalling are 4EBP1 and S6K, which act as regulators of translation initiation. The inhibition of mTORC-dependent translation has been shown to extend life span and to confer protection against a growing list of age-related pathologies [19]. Indeed, Selman et al. observed that mice lacking S6 kinase, one of the main targets of mTORC1 and one of the key enzyme regulating protein translation, present a prolonged lifespan along with an increased activity of AMPK [19]. This observation resembles what was observed in the caloric-restriction-induced extension of lifespan. Interestingly, S6 kinase null mice did not present any reduction in protein synthesis.

However, the control of translation by mTORC1 may be directly involved in the aging process. Indeed, one of the most common molecular signatures of aging is represented by an accumulation of altered proteins, derived from either an erroneous synthesis or an incorrect post-translational modification. Interestingly, in cells with hyperactive mTORC1 signalling, the consequent increase of elongation speed is accompanied by a significant alteration of protein folding [20].

## 3. Mitochondrial and Oxidative Stress

Mitochondria, through oxidative phosphorylation, generate adenosine triphosphate (ATP) and act, therefore, as the powerhouse necessary for cell survival [21]. The mitochondrial regulation occurs mainly by peroxisome proliferator-activated receptor-*γ* co-activators *α* and *β* (PGC-1*α* and PGC-1*β*, respectively), which responds to changes in nutrient and energetic status as indicated by AMP/ATP intracellular concentrations. Therefore, the expression of PGC-1*α*/*β*, play a fundamental role in mitochondrial biogenesis [21]. The mitochondria represent the main source of reactive oxygen species (ROS), which are essential mediators of the aging process. In this context ROS are mainly generated by different NADPH oxidase (Nox) isoforms. The major defence system against deleterious ROS effects is Nuclear factor erythroid-related factor 2 (Nrf-2), a transcription factor that regulates the translation of anti-oxidant enzymes such as superoxide dismutase (SOD), catalase (CAT), glutathione peroxidase (GPx). In conditions characterized by increased ROS production, the Nrf-2 activity might be inhibited with a subsequent ROS-induced rise in TNF-*α* levels, which in turn can up regulate Nox complex expression and activity, generating a vicious cycle [22]. Organs such as heart, with a limiting rate of replication and high levels of oxygen consumption, are particularly sensitive to this phenomenon, which explains the detrimental cardiovascular consequences of aging [23]. In this scenario, mTOR is considered an important regulator of oxidative stress by promoting mitochondrial biogenesis and oxidative metabolism through PGC-1*α* pathway [24].

## 4. Inflammation

Aging is characterized by a systemic increase of pro-inflammatory mediators, a phenomenon known as inflammaging [19,25]. Senescent cells, indeed, might release an array of pro-inflammatory substances (SASP) including TNF-*α*, IL6, and IL1 [19,25]. These pro-inflammatory cytokines are regulated by transcription factors sensitive to redox potential through mTOR signalling [20]. The inflammaging process leads to an increased expression of proteins such as matrix-metalloproteinase (MMP9), intercellular adhesion molecule-1 (ICAM-1), vascular cell adhesion molecule-1 (VCAM-1), related to the endothelial damage, vascular smooth muscle cell (VSMC) proliferation, and matrix remodelling [25].

## 5. Stem Cell Pool Turnover

mTOR is a key player in the activation of tissue stem cells promoting their proliferation and their exit from the characteristic quiescence status. Stem cell activation contributes to tissue turnover and repair, but in the long run progressively leads to stem cells niche depletion. Thus, mTOR inhibition might represent an effective approach to preserve the stem cell pool and, consequently, to maintain the ability to repair tissue damages [26].

## 6. Autophagy

Activation of autophagy is another peculiar key mTORC1-regulated process that plays a pivotal role in supporting longevity. Autophagy is a lysosome degradation process demonstrated to be essential for the maintenance of cell homeostasis and stress response induced by low levels of growth factors, nutrient, including amino acids and glucose, or cellular levels of ATP [27]. This process is instrumental in the recycling of organelles, including mitochondria, endoplasmic reticulum, and peroxisomes. mTOR is well-known to inhibit autophagy, promoting the accumulation of protein aggregates and degenerated mitochondria leading to a deep dysregulation of mitochondrial functions. These events contribute to the pathogenesis of age-related disorders such as cardiovascular disease. Indeed, cardiac aging is characterized by a progressive decline of mitochondrial cardiomyocyte function [28] with a subsequent reduced ATP production and increased ROS formation [29]. There is a consistent body of evidence suggesting that, in this scenario, autophagy might play a critical role in maintaining cardiac structure and function reversing the accumulation of damaged mitochondria induced by aging. Indeed, in situations of low energy and reduced ATP levels, AMPK another cellular energy sensor, is activated and directly inhibits mTOR by its direct phosphorylation, stimulating autophagy [30].

## 7. Cellular Senescence

The mTOR signalling drives cellular senescence, a phase of decline of cellular functions that follows development and maturation. This period of cell life is characterized by a loss of replicative ability without activation of the apoptosis process [31]. In proliferating cells, growth-promoting pathways, including those regulated by mTOR are activated. This ensures that cellular mass growth is balanced by cell division. In the absence of growth factors, the MAPK/mTOR pathway is deactivated, cell cycle freezes and cells become quiescent with the subsequent reduction of metabolism, protein synthesis and cellular functions [31]. The quiescent cell does not grow and replicate but retains its proliferative potential so that a restoration of the nutritional state or the addition of growth factors may re-activate the growth and replication processes. Therefore, the cell comes out of its quiescence status, grows, and divides. The deregulation of several pathways including autophagy, mitochondrial oxidative phosphorylation as well as the presence of oxidative stress or systemic inflammation might cause a cell cycle block along an active mTOR signal [31]. All the above mentioned conditions may, indeed, cause, directly or indirectly, a cyclin-dependent kinase (CDK) inhibition resulting in the arrest of the cell cycle in the G1 phase, while leaving the cells metabolically activated by growth-signalling pathways such as mTOR [31,32]. The mTOR activation, when the cell cycle is blocked, leads the cell to a senescence state [31]. These cells become hypertrophic and show hyper-differentiation and a pro-inflammatory state. Moreover, they lose their ability to restart proliferation, having a wide and flattened morphology. The loss of mitotic competence is one of signs of the senescent phenotype [31,32]. The irreversible cell cycle arrest in the G1 phase contributes to cell injury, decreasing physiological cell capacities, leading to an unnatural aging process. This event, defined geroconversion, might also promote neoplastic cell transformation [33]. Conditions that activate mTOR accelerate geroconversion. In this perspective, different genes were identified that by activating the mTOR pathway, behave like oncogenes that promote cellular mass growth in cancer and senescence. Conversely, p53 a tumour suppressor, by inhibiting mTOR slows down and suppresses geroconversion [33]. Thus the inhibition of mTOR might prevent age-related neoplastic disease (Figure 1).

## 8. Aging, mTOR, and Cardiovascular Diseases

Aging is a major independent risk factor for cardiovascular disease, the leading cause of morbidity and mortality worldwide [34,35]. Cardiovascular aging is a quite complicated pathophysiological process associated with a variety of biological adaptive responses, including progressive myocardial remodelling and declined cardiac reserve [36,37]. Moreover, aging is also related with an increased cardiomyocyte apoptosis. Besides the mentioned physiological changes, many factors have been identified to play a role in the increased incidence of cardiovascular diseases with aging. Sympathetic over-activation, hypertension, dyslipidaemia, oxidative stress, and low-grade chronic inflammation may all play a role in compromising cardiac and vascular structure and function [35,38,39]. To maintain physiological contractile function of the heart, cardiomyocyte have to rely on a constant supply of high-energy phosphates from mitochondria [38]. Aging is associated with a significant decline in mitochondrial function, and, thus, in ATP supply, a reduction in mitochondrial number and in the presence of functionally compromised enlarged mitochondria [40]. In this scenario mitochondria are also the primary source of ROS, which triggers mitochondrial-derived (endogenous) apoptosis, recognized as a leading cause of cell death in aging cardiomyocytes [41,42].

Telomere shortening also contributes to cellular senescence in replicating cells, while defective telomere in cardiac stem cells has been identified as a useful marker of cardiac aging [43,44].

Another important point in the pathogenesis of cardiovascular aging is autophagy. It is a highly conserved cyto-protective, rather than self-destructive, process involving degradation and recycling of intracellular organelles and proteins [45,46]. A variety of cellular stress conditions such as nutrient or growth factor deprivation, ROS accumulation, aggregation of the long-lived or damaged proteins and organelles and hypoxia usually turn on autophagy. Autophagy plays a pivotal role in survival and longevity of cardiomyocytes. These cells, indeed, rely on the autophagic process to remove and recycle unwanted, damaged or long-lived cellular components or proteins in order to maintain their homeostasis. Autophagy has a protective function and meets a metabolic requirement during conditions of energy/nutrient deficit, such as starvation, ischemia reperfusion and heart failure [47]. It has been shown that autophagy within the cardiovascular system declines with aging, and loss of autophagy was demonstrated to accelerate cardiovascular aging [48,49,50,51].

The deregulation of several pathways including autophagy, mitochondrial dysfunction, telomere shortening, oxidative stress, systemic inflammation and metabolic dysfunction leads the cells to a state of senescence that seems to driving towards age-related cardiovascular diseases [52,53,54,55]. The increase production of pro-inflammatory agents by senescent cells, a phenomenon known as inflammaging, is associated with endothelial damage, VSMC proliferation and vascular matrix remodelling, three key steps in the development of atherosclerosis [25,56,57].

Cardiovascular aging makes heart and vessels more susceptible to any stress condition [58]. This dysfunctional cardiovascular phenotype is characterized by several hemodynamic changes including increased arterial resistance, oscillatory shear stress and artery stiffness, leading to hypertension and atherosclerosis. This process appears to be independent by traditional cardiovascular risk factors, supporting the idea that these alterations are, indeed, mainly due to age [37].

mTOR is up-regulated by nutrients, including phosphate, indispensable for energy metabolism. The phosphate is also known to induce vascular calcification through the mTOR signalling, driving osteogenic trans-differentiation of VSMCs [59]. In addition, mTOR activation in VSMCs seems to reduce significantly klotho expression [59]. Klotho is a protein playing a key role in the regulation of aging, in particular at the cardiovascular level. The functions of this protein include inhibition of local phosphate uptake into VSMCs, suppression of osteoblast-like differentiation of VSMCs, attenuation of matrix mineralization, and also preservation of endothelial function [60,61]. In this perspective, mTOR-inhibitors can be regarded as klotho up-regulators, therefore potentially preventing or delaying the onset of age-related cardiovascular dysfunctions [60].

## 9. Aging, mTOR, and the Immune System

The aging immune system presents a decreased efficiency, responsible of an increased susceptibility to infections. The aging process is, indeed, characterized by the alteration of several mechanisms designed to maintain equilibrium including DNA repair, protein synthesis, apoptosis inevitably leading to a severe dysfunction of systemic response of immune activation and inflammation and a subsequent reduction in the host defence against pathogens and external insults [62].

The decreased immune-response observed during age is also known as immunosenescence and it is characterized by profound changes in the innate and adaptive immune response, including variations in T and B cells compartments and in quantitative and qualitative alterations in antigen-presenting cells and immune effector cells function [63].

The contributing factors to immunosenescence are numerous and diverse, due to the multi-factorial complexity of the immune system. It is often problematic to conclude whether changes in a particular cell type are intrinsic to that cell, or caused by environmental changes, or both. This is particularly the case for lymphocytes, where the cellular interplay between different cells’ subsets is crucial for effective responses. Thus, if one subset is affected it will inexorably change the function of others [64,65].

Composition and property of mature lymphocyte pool is deeply influenced by aging. Dysregulation in T-cell response is primarily due to the thymic involution that occurs with aging and it is a physiological process characterized by the lowering of naïve T cells associated with an increased number of memory T cells and a reduced immune reactivity [66]. In particular, CD8+T cells exhibit telomere loss that can be displayed during antigen-driven proliferative responses and that results in growth arrest [67,68] and a reduction of CD28+ expression after several cycles of antigenic stimulation. The reduced expression of CD28, in consideration of its role in a variety of cellular processes, including co-stimulatory signal, m-RNA stabilization and glucose metabolism, is associated with a destabilization of T cell response. Immunosenescence is also characterized by a reduced expression of anti-viral cytokines, including interferon alpha, and an increase production of pro-inflammatory mediators that explain the loss of an efficient response to vaccines and antigens stimulation [67]. McKay et al. observed a more general decline of T cell response, caused by a wider down-regulation of co-stimulating signals [68].

During immunosenescence the quality of the antibody response is substantially impaired. Indeed, Chong et al. demonstrated that B cell compartment is affected by aging with a deprived B-lymphopoiesis process along with a progressive accumulation of mutations in the gene encoding for immunoglobulin (Ig) heavy chains [69], leading to a significantly worse antibody response. In humans, B cells pool decreases with age with a lowering response to environmental antigens, although different observations suggest an increase of peripheral blood naïve B cells in the elderly [70]. The most likely cause of this B cell failure is a lack of effective T cell help as a consequence of the thymic involution and the subsequent functional decline of T cells. However, there are T-independent functions of B cells, such as the polysaccharide responses that are crucial for anti-bacterial protection, which also appear to be lacking in aging [70]. In addition, since it is well known that B cells can act as antigen presenting cells, regulating T cells development, it is conceivable that some of the observed failures of T cell function may be, at least partially, blamed on an insufficient help from B cells. Changes in B cell number and repertoire have been described, and decreased IgM and IgD levels suggest a shift from the naïve (CD27−) towards the memory (CD27+) compartment of the B cell branch [71,72,73].

Dendritic cells (DCs) are the main professional antigen-presenting cells and represent the link between innate and adaptive immune system. Several different aspects of DCs functions undergo changes due to aging, which involve both differentiation and phenotype. Indeed, the absolute number of myeloid DCs and their precursors is significantly reduced in the aging process, with a concomitant decrease of their phagocytic and migratory ability [70].

Neutrophils play an important role in innate immunity and in the defence against invading pathogens. Once recruited within the inflamed tissue, neutrophils eliminate microbes and cellular debris through phagocytosis and intracellular killing, two mechanisms that result in ROS generation and release of several pro-inflammatory mediators, including TNF-α, VEGF, IL-12, IL-8, MIP-1α, and BLyS [70,74]. Despite the unchanged number of circulating neutrophils in the aging process, these cells manifest a minor bactericidal and phagocytic activity, caused by several alterations, such as response to the stimulation by GM-CSF on the activation of the cellular respiratory burst, with consequent reduction of ROS generation [70]. On the other hand, basal levels of Toll like receptor (TLR) signalling pathway components (TLR2, TLR4) are not modified, but aging is characterized by alteration in innate-immune signalling pathway, including a decrease of signalling via the TLR4-MyD88-dependent pathway [70].

Macrophages play a fundamental role in the response to innate immunity and ensure adequate communication between innate and adaptive immunity. They engulf pathogens and destroy them by oxygen dependent or independent mechanisms [70,75]. Immunosenescence of this cell type is characterized by the deficiency in chemotaxis and phagocytic capacity, and in a reduction in the ability to release pro-inflammatory cytokines and chemokines [68]. Although there is no reduction in their absolute number, aging is associated with a change in the macrophage phenotype, including a lower expression of CD68, which allows macrophages to home specific targets [68,76]. The expression of Major histocompatibility complex class II (MHC-II) and the presentation of antigen by macrophages seem to be reduced in animal and human studies [76,77,78,79], and this reduction is associated with a decline of interferon-gamma (IFN-γ) activity, which plays a pivotal role in macrophages activation. In addition, macrophages immunosenescence is associated with a decrease in IFN-γ-induced superoxide anion production, a reduced expression of inducible nitric oxide synthase (iNOS) and an important impairment of phagocytosis and clearance of infectious organisms [75,76]. These changes are due to a compromised IFN-γ-dependent activation despite a normal expression of IFN-γ receptors.

mTOR was identified as a central integrator of cell metabolism that drives lineage specific functions in the T cell compartment [80]. In order to support cell proliferation, mTORC1 promotes the transcription of genes involved in glycolysis, the pentose phosphate pathway (PPP) and de novo lipogenesis [81]. It has been generally assumed that mTORC1 signalling increases flux through the oxidative PPP to generate NADPH, which is needed to ensure reducing power and for many biosynthetic processes [81]. Earlier studies suggest that myc- and mTORC1-dependent activation of T cells involves dramatic up-regulation of glucose consumption via PPP [82].

mTOR-inhibition induce T-cell anergy even in presence of an adequate co-stimulation [83]. This is due to the down-regulation of the metabolic machinery implicated in the full T-cell effector function [84] and induces sequestration of activated T cells in lymphoid tissues [85,86]. Other studies [80,87] found that mTORC1 inhibition in CD4+T cells impairs Th1 and Th17 cell differentiation without affecting Th2 cell generation, whereas mTORC2-deficient T cells fail to differentiate into Th2 cells, but retain their ability to become Th1 and Th17 cells.

mTOR-inhibitor may also influence the differentiation of T regulatory cells (Tregs), a group of T lymphocytes capable to inhibit most types of immune response [88,89]. mTOR inhibition can both expand naturally occurring Tregs (CD4+CD25+FoxP3+) and induce adaptive Treg cells from conventional CD4+ T cells [90,91,92,93], while inhibiting the proliferation of non-suppressive, activated T cells (CD4+CD25low) and making these cells more susceptible to apoptosis [92]. Additionally, mTOR inhibition may have a specific effect on CD8+ T cells, inducing the proliferation of their memory subset as demonstrated by Araki et al. [94] in a mice model of acute lymphocytic choriomeningitis virus (LCMV) infection. Moreover, these cells showed higher levels of CD127 (IL-7 receptor α and essential for memory T-cell maintenance), CD62L (lymph node homing receptor and associated with high proliferative capacity) and the anti-apoptotic B-cell lymphoma 2 (Bcl2) (expressed at high levels in memory T cells), all markers featuring self-renewing memory cells [94]. These effects could be due to the ability of mTOR inhibition to reduce the expression level of T-bet (transcription-factor regulating genes involved in Th1 differentiation), which promotes the generation of memory T cells [94,95].

mTOR-inhibition may also have significant effects on DCs [96]. Rapamycin prevents phenotypic and functional maturation of DCs induced by IL-4, LPS or CD40 ligation [97,98] and impairs their development caused by fms-like tyrosine 3 kinase ligand (Flt3L), a potent endogenous DC growth factor [99,100]. In addition, mTOR-inhibition induces apoptosis in both human monocyte-derived and CD34+-derived DCs, but not other immune cells such as macrophages or myeloid cell lines [101]. mTOR inhibition enhances CCR7 expression on DC, increasing their migration into secondary lymphoid tissue [102]. This property is crucial for the tolerogenic effects attributed to mTOR inhibitors, because it allows these cells to reach appropriate T cell areas in the lymphoid tissue [81]. Furthermore, mTOR inhibition regulates plasmacytoid DCs (pDC). These cells play a key role in the anti-viral immune response since they can rapidly produce large quantities of type I IFN during viral infection and activate NK cells [103]. Moreover, the production of IFN along with IL-12 might support the effector functions of CD8+ T cells as well as the polarization of CD4+ T cells toward a Th1 bias [103]. At the same time mTOR inhibitors can modify the functional features of antigen presenting cells inducing the up-regulation of ILT3 and ILT4 together with a reduction of CD40 expression on their surface. These changes are associated with an increase in circulating CD8+CD28−T cells and CD4+CD25+Foxp3+CTLA4+ Tregs [104].

Myeloid derived suppressor cells (MDSC) are a group of myeloid cells including precursors of macrophages, DCs, granulocytes and myeloid cells at early stage of differentiation with immune suppressive activity [105,106,107,108]. These cells have been shown to regulate the immune response in cancer [105,109,110], bacterial infections, acute or chronic inflammation, traumatic stress, autoimmune disease and transplantation [111,112,113] by a complex biological machinery. In particular, MDSCs, expressing high levels of arginase 1 and iNOS, may cause a substantial depletion of L-arginine in the microenvironment with a consequent reduction of T cell proliferative ability [114]. Moreover, NO released by iNOS suppresses T-cell function inhibiting the JAK3/STAT5 signalling pathway [115,116]. In addition, because of arginine depletion, iNOS may preferentially produce superoxide that, together with NO, may generate peroxynitrite resulting in the nitration of the T cell receptor with the consequent impairment the antigen presentation process [117]. Other immunesuppressive/tolerogenic mechanisms induced by these cells involve heme oxygenase-1 (HO-1), TGF-β and indoleamine 2,3-dioxygenase (IDO) [118,119,120]. It is well-known that the aging process is characterized by a substantial increase in the presence of MDSCs not only in the bone marrow and in the circulation, but also in primary and secondary lymphoid organs [121].

The influence of mTOR inhibitors on MDSCs function has been clarified by Wu et al. [122] in skin-allograft and tumour bearing mice models. In both models, mTOR inhibition reduced the percentages and number of MDSCs and hampered their immunosuppressive effects. The reduction in MDSCs-induced immunosuppression seems to be due to the inhibition of iNOS and arginase activities. Finally, mTOR inhibitors might also decrease MDSC differentiation from myeloid progenitors by blocking the glycolysis, an essential metabolic pathway in the development of these cells [122].

In conclusion, mTOR activation might significantly impair immune response. Interestingly, a recent phase 2a randomized, placebo-controlled clinical trial confirmed this hypothesis, demonstrating that the use of a low-dose combination of a catalytic (BEZ235) plus an allosteric (RAD001) mTOR inhibitor in an elderly population was associated with a significant decrease in the observed infection rate [123].

## 10. Aging, mTOR, and Cancer

Several molecular mechanisms are responsible of cellular senescence. It has been hypothesized that aging could be the result of the progressive addition of molecular damages to a favourable genetic background [124,125,126,127,128,129,130]. A growing body of evidence supports the pre-determination of cellular senescence [124,125,126,127,128,129,130]. In this setting, one of the most relevant factors affecting cell aging is telomere shortening through multiple cell cycles. In fact, telomerase, a specific enzyme preserving telomeres length, is poorly expressed in most human non-dividing cells and its levels decline with aging. Since telomeres protect chromosomal DNA from damages activating programmed cell death pathways (apoptosis), their shortening is thought to function as a “sensor” of cells age [124,125,126,127,128,129,130]. Not surprisingly, neoplastic cells express abnormal levels of telomerase, which allows the maintenance of telomere’s length even after repeated replications. This potentially results in tumour “immortality”. In addition, there are several molecular mechanisms underlying cell senescence relevant for oncogenesis that are telomere-independent. Environmental modifications, including increments of oxygen concentration and free oxygen radicals or DNA damage, can induce the aging process independently from telomere length [32,131,132]. Although aimed to prevent auto-immunity, tolerance can be directed towards non-self-antigens, following a process called induced tolerance. Cancer originates from the transformation of the host cells: during this process, multiple accumulating mutations turn the self into a non-self. This change is expected to prompt a response from the immune system, but neoplasia can use an array of mechanisms of immune-evasion. The cancer cells often display weak immunogenicity, especially due to the lack or low expression of co-stimulatory molecules or inefficient antigen presentation ability. Thus, potential tumour reactive T-cells are induced to mount ineffectual immune responses or even to develop anergy. At advanced stages of carcinogenesis, the immune system applies a selective pressure on the genetically unstable tumour cells: those able to resist to or suppress the immune response are selected and represent the first cause of immune escape. Later in the process of tumour progression, inefficient immune responses can even help tumour growth [133]. It is obvious that tumour cells removal [133] represents a critical factor regulating carcinogenesis, and immunosenescence plays a crucial role in supporting cancer growth. This multifaceted sequence of events determines that tumour cell variants can survive and enter an equilibrium phase, followed by an uncontrolled tumour expansion.

Another important mechanism potentially linking senescence with neoplasia is inflammation [128]. The association between chronic inflammation and cancer has been extensively described in different organs: inflammatory bowel diseases and colorectal cancer, chronic gastritis, and gastric adenocarcinoma, and chronic viral hepatitis with liver cancer [133,134].

Several mechanisms have been described to validate the relationship between inflammation and carcinogenesis [133,134]. Leukocytes generate reactive oxygen and nitrogen species, causing local tissue alterations and DNA damage; enhanced proliferative signals mediated by cytokines released by inflammatory cells increase the risk of mutations; alterations in epigenetic mechanisms modify gene expression patterns [133]. Inflammatory mediators influence the function of suppressor cell populations, particularly MDSCs and Tregs, causing the inhibition of CD4+ and CD8+ T-cell proliferation, blocking NK cell activation, limiting DC maturation, and to polarize immunity towards a Th2 response [134]. Finally, the inflammatory environment favours angiogenesis and cancer growth and create a privileged niche supporting cancer stem cells [134].

Increased activation of mTORC1 is observed in several human neoplasia due to gain-of- function mutations in different oncogenes, including PI3 kinase, AKT or ras, and/or loss-of-function mutations in a variety of tumour suppressor genes, such as PTEN, LKB1 or TSC1/2, among the upstream regulators of mTORC1. These mutations provide a selective growth advantage to cancer cells in comparison to normal cells [135]. In order to meet the high demands of proliferation, neoplastic cells have essential modifications in nutrient uptake and energy metabolism, both processes under the direct control of the mTORC1 pathway. Accordingly, in addition to drive protein synthesis, oncogenic activation of mTORC1 promotes a gene expression platform that is involved in metabolic reprogramming. Activation of mTORC1 promotes glycolysis via up-regulation of Hypoxia-inducible factor alpha (HIF1α) and c-Myc; stimulates lipid biosynthesis and the PPP through sterol regulatory element binding protein 1 (SREBP-1) [136], and induces glutamine metabolism by SIRT4 suppression [137]. Thus, drugs selectively targeting mTORC1 can significantly impair cancer metabolism. Indeed, the role of mTOR in carcinogenesis is linked to multiple layers of metabolic regulatory networks that activate glycolysis via increasing the expression of pyruvate kinase [138]. This enzyme is the main player in the pathogenesis of the Warburg effect, a well-known metabolic feature of neoplasia that rely on glycolysis rather than mitochondrial respiration [139]. This metabolic conversion permits cancer cells to survive under hypoxia through reliance on anaerobic glycolysis, which involves the transcription factor HIF1α [140,141]. mTORC1 also stimulates carcinogenesis by preventing physiological protein turnover via autophagy. This occurs at the step of autophagosome formation, supressed by mTORC1 through AMBRA1 phosphorylation [142]. Thus, mTORC1 stimulates oncogenesis by both inhibiting glycolysis and promoting autophagy. On the other hand, mTORC2 is crucial for cell survival, its activation is a critical step in neoplastic transformation of cells lacking PTEN [143]. Meta-analyses indicate that mTOR blockade reduces the incidence of several neoplasia [144]. The efficacy of such treatment may depend on the relative involvement of mTORC1 and mTORC2.

## 11. Conclusions

Pharmacological interventions to slow aging are often considered as the Holy Grail of medicine, although the increasing knowledge of cellular and molecular mechanisms of aging are taking us closer to reach it. The evidence accumulated in the last decade suggests that the primary target for such interventions might be the nutrient response pathway controlled by mTOR. Indeed, the use of mTor inhibitors has been shown to prolong lifespan in several animal models and to confer protection against most of the major age-related diseases. Several drugs in this class, including sirolimus and everolimus, are already clinically approved and currently used in different setting, and others are under development. Although the presence of side effects currently precludes their use in otherwise healthy individuals, mTOR inhibitors might become widely used to slow aging and reduce age-related diseases. However, playing with mTOR, a central switch of cellular metabolism, might be at the same time valuable and dangerous. Even if mTOR inhibitors have been used in the clinical setting for more than ten years, we have not still learned how to take advantage of their advantageous actions without paying an excessive fee for their side effects. Finally, their clinical use might shed light on some key biological questions on the link between metabolism, immune response, and aging.

## Figures and Tables

**Figure 1 ijms-20-02774-f001:**
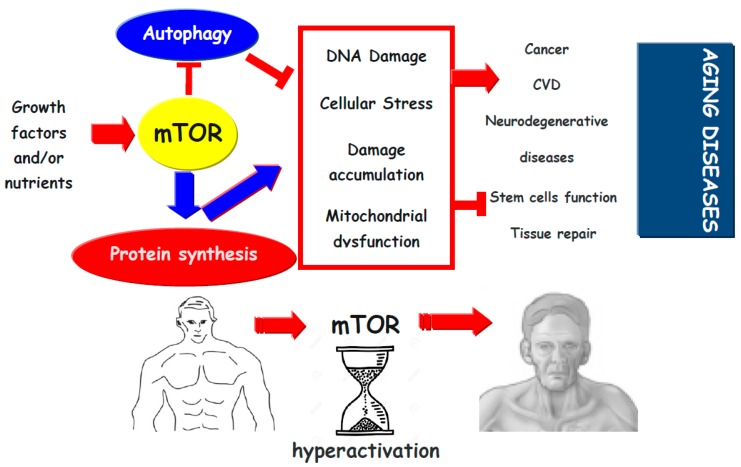
Growth factors and nutrients by activation of mTOR inhibits autophagy and promotes protein synthesis. This may promote cellular stress (protein aggregation, organelle dysfunction, and oxidative stress), which might lead to damage accumulation, reduction in cell function and induces stem cell exhaustion, which reduces tissue repair and promotes tissue dysfunction. Thus, mTor activation pathway promotes the development of aging diseases. 
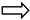
 (blue and red) indicates activation, 
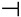
 indicates inhibition.

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
