# Peer review of "mTOR and Aging: An Old Fashioned Dress"

_ijms, 2019, doi:10.3390/ijms20112774_

Round 1
Reviewer 1 Report
- Some English editing is necessary:
l.52: change "play a crucial role" to "plays"
l.82: change "senescent cell" to "cells"
l.97: change "play a pivotal role" to "plays"
l.333: change "ageing" to "aging"
l.366: change "mechanisms has been" to "have"
I would reduce the use of " " through the text ("aging", "normal disease", "growth factors" (what does this even mean?), "powerhouse"...)
(There are more)
- Some references are missing, for example:
l. 37-41
l. 81-82
l. 334-337
l. 379-381
- Be more specific when writing, say what yo want to say, for example:
51-52: plays a crucial role in metabolic control. How?
55-60: Protein translation:
I would understand that mTOR is necessary for the synthesis of proteins that promote cell proliferation and survival. But then the inhibition of mTORC would not extend life span. What does this paragraph really want to say? can it be rewritten?
- I do not see how conclusions relate to the body of the review.
Author Response
REFEREE 1
Comments and Suggestions for Authors
1. - Some English editing is necessary:
l.52: change "play a crucial role" to "plays"
l.82: change "senescent cell" to "cells"
l.97: change "play a pivotal role" to "plays"
l.333: change "ageing" to "aging"
l.366: change "mechanisms has been" to "have"
We modified the text as suggested
2. I would reduce the use of " through the text ("aging", "normal disease", "growth factors" (what does this even mean?), "powerhouse"...) (There are more)
We modified the text as suggested
3. - Some references are missing, for example:
l. 37-41
l. 81-82
l. 334-337
l. 379-381
We revised the references throughout the manuscript
4. - Be more specific when writing, say what yo want to say, for example:
51-52: plays a crucial role in metabolic control. How?
55-60: Protein translation:
5. I would understand that mTOR is necessary for the synthesis of proteins that promote cell proliferation and survival. But then the inhibition of mTORC would not extend life span. What does this paragraph really want to say? can it be rewritten?
We rewrote both paragraphs in the attempt to be more specific.
6. - I do not see how conclusions relate to the body of the review.
We rewrote the conclusions in the attempt to better reflect the body of the review.

Reviewer 2 Report
This reads as a well-structured and logically presented review covering the role of mTOR in a number of age-related conditions. Apart from some minor typos (detailed below), the first few sections read well and generally cover the essential points. There are a few areas where an additional study or two could be included, as I have highlighted below. My major concern is the detection of plagiarism which must be addressed.
Major Points:
1. Plagiarism: Lines 400-406 in the Conclusion section are taken virtually word-for-word (with some minor changes) from the Abstract of Johnson, Rabinovitch and Kaeberlein (2013) Nature (doi 10.1038/nature11861). This is not acceptable and needs rewritten.
Having detected this, I also looked at the Abstract of this manuscript, and while not as directly copied as the conclusion, it does resemble the start of the Lopez-Otin et al, 2013 Review ‘The Hallmarks of Aging’ in Cell.
It also makes me query whether there is plagiarism anywhere else in the article, and I suggest this is checked for.
2. In Section 1 (Introduction), where mTOR is described at the end, some brief background to the studies which indicated that mTOR had a pivotal role in longevity in various animal models would strengthen the links between mTOR and aging (beyond the current mention of caloric restriction).
3. The immunity section covers a lot of ground, describing how different immune compartments respond to aging and highlighting the role of mTOR in driving lineage specific functions in the T cell compartment. However, the mTOR inhibition examples are not specifically age related. Therefore, this section would benefit from a bit more detail about how mTOR is dysregulated with age and therefore impacts immunity in the elderly. I appreciate studies in this area are limited, but it might be useful to include examples of where altering mTOR activity in the elderly has improved immune function, e.g. Mannick et al 2014 Science Translational Medicine. Indeed, a recently reported phase 2a trial indicated TORC1 inhibition reduced infections in the elderly (Mannick et al 2018 Sci Transl Med).
Minor points:
1. The meaning behind the title is not clear to me – I don’t get the reference to ‘a fashionable old dress’. You may want to reconsider this.
2. The paper generally reads well, though there are some typos. The authors should check through their work once any corrections have been made, but some examples to look out for include:
Line 33: “ in agreement with those who consider aging as …..”
Line 42: Needs another ‘kinase’ inserted - phosphoinositide kinase-related kinase family
Line 45: PI3-Kinase (spelling)
Line 56: No apostrophe in ribosomes
Line 73: ‘rise’ not raising
Line 94: tissue damage
Line 173: would read better as “The deregulation of several pathways including autophagy, mitochondrial dysfunction….”
Author Response
REFEREE 2
Comments and Suggestions for Authors
This reads as a well-structured and logically presented review covering the role of mTOR in a number of age-related conditions. Apart from some minor typos (detailed below), the first few sections read well and generally cover the essential points. There are a few areas where an additional study or two could be included, as I have highlighted below. My major concern is the detection of plagiarism, which must be addressed.
Major Points:
1. Plagiarism: Lines 400-406 in the Conclusion section are taken virtually word-for-word (with some minor changes) from the Abstract of Johnson, Rabinovitch and Kaeberlein (2013) Nature (doi 10.1038/nature11861). This is not acceptable and needs rewritten.
Having detected this, I also looked at the Abstract of this manuscript, and while not as directly copied as the conclusion, it does resemble the start of the Lopez-Otin et al, 2013 Review ‘The Hallmarks of Aging’ in Cell.
It also makes me query whether there is plagiarism anywhere else in the article, and I suggest this is checked for.
We checked the whole manuscript for potential plagiarisms and rewrote any suspicious paragraph.
2. In Section 1 (Introduction), where mTOR is described at the end, some brief background to the studies which indicated that mTOR had a pivotal role in longevity in various animal models would strengthen the links between mTOR and aging (beyond the current mention of caloric restriction).
We included a mention to the studies indicating a role for mTOR in longevity.
3. The immunity section covers a lot of ground, describing how different immune compartments respond to aging and highlighting the role of mTOR in driving lineage specific functions in the T cell compartment. However, the mTOR inhibition examples are not specifically age related. Therefore, this section would benefit from a bit more detail about how mTOR is dysregulated with age and therefore impacts immunity in the elderly. I appreciate studies in this area are limited, but it might be useful to include examples of where altering mTOR activity in the elderly has improved immune function, e.g. Mannick et al 2014 Science Translational Medicine. Indeed, a recently reported phase 2a trial indicated TORC1 inhibition reduced infections in the elderly (Mannick et al 2018 Sci Transl Med).
We modified the paragraph as suggested.
Minor points:
1. The meaning behind the title is not clear to me – I don’t get the reference to ‘a fashionable old dress’. You may want to reconsider this.
We modified the title
2. The paper generally reads well, though there are some typos. The authors should check through their work once any corrections have been made, but some examples to look out for include:
Line 33: “ in agreement with those who consider aging as …..”
Line 42: Needs another ‘kinase’ inserted - phosphoinositide kinase-related kinase family
Line 45: PI3-Kinase (spelling)
Line 56: No apostrophe in ribosomes
Line 73: ‘rise’ not raising
Line 94: tissue damage
Line 173: would read better as “The deregulation of several pathways including autophagy, mitochondrial dysfunction….”
We modified the text as suggested and we accurately revised the manuscript for typos.
Round 2
Reviewer 2 Report
The authors have generally addressed my previous comments.
The abstract and conclusion no longer raise any plagiarism concerns.
I only have a couple of minor points that I would suggest are looked at prior to publication:
1. In the conclusion, it should be holy grail (spelling)
2. Following the changes made to the conclusion, one sentence now reads "The evidence accumulated in the last decade clearly suggests that the primary target for such interventions might be the nutrient response pathway controlled by mTOR." I think the use of might be does not show that clearly mTOR is involved. I would consider rewording this sentence.
3. A minor typo still exists in lines 45-46 - Still need to add the extra 'kinase' i.e. phosphoinositide kinase-related kinase family
4. I still don't really understand the choice of title, but maybe that is just me!
Author Response
We modified the text as indicated. If the reviewer would suggest a different title we are ready to consider it.